# A Comprehensive Oncological Biomarker Framework Guiding Precision Medicine

**DOI:** 10.3390/biom15091304

**Published:** 2025-09-10

**Authors:** Reza Bayat Mokhtari, Manpreet Sambi, Faezeh Shekari, Kosar Satari, Roya Ghafoury, Neda Ashayeri, Paige Eversole, Narges Baluch, William W. Harless, Lucia Anna Muscarella, Herman Yeger, Bikul Das, Myron R. Szewczuk, Sayan Chakraborty

**Affiliations:** 1Department of Pharmacology and Therapeutics, Roswell Park Comprehensive Cancer Center, 265 Elm and Carlton Streets, Buffalo, NY 14263, USA; reza.bayatmokhtari@roswellpark.org (R.B.M.); paige.eversole@roswellpark.org (P.E.); 2Program of Developmental Therapeutics, Roswell Park Comprehensive Cancer Center, Buffalo, NY 14263, USA; 3Department of Biomedical and Molecular Sciences, Queen’s University, Kingston, ON K7L 3N6, Canada; m.sambi@queensu.ca (M.S.); faezehshekari@gmail.com (F.S.); sattarikosar99@gmail.com (K.S.); royaghafoury@gmail.com (R.G.); neda.ashayer@gmail.com (N.A.); 4Allergy and Clinical Immunology Division, Rady Children’s Hospital, University of California, San Diego, CA 92123, USA; nbaluch@ucsd.edu; 5Hematology and Oncology Division, Charleston Area Medical Center, Charleston, WV 25304, USA; wharless@encyt.net; 6Laboratory of Oncology, Fondazione IRCCS Casa Sollievo della Sofferenza, Viale Cappuccini, 71013 San Giovanni Rotondo, Italy; l.muscarella@operapadrepio.it; 7Department of Laboratory Medicine & Pathobiology, University of Toronto, Toronto, ON M5G 0A4, Canada; hermie@sickkids.ca; 8KaviKrishna Laboratory, Department of Cancer and Stem Cell Biology, Indian Institute of Technology Guwahati Research Park, Guwahati 781039, India; bikul.das@gmail.com; 9Thoreau Laboratory for Global Health, Department of Experimental Therapeutics, University of Massachusetts, Lowell, MA 01852, USA

**Keywords:** comprehensive oncological biomarker framework, cancer immunotherapy, biomarkers, tumor microenvironment, precision medicine, gut microbiome

## Abstract

Cancer remains a major cause of mortality worldwide, driving ongoing innovation in therapeutic strategies. Immunotherapy has transformed cancer care by leveraging the immune system to target tumors, but its effectiveness is limited by tumor heterogeneity, immune resistance, and unpredictable toxicities. Moreover, the absence of robust biomarkers to predict therapeutic response and manage adverse effects remains a significant challenge. Recent advances in biomarker discovery, including liquid biopsy technologies and gut microbiota profiling, are enhancing the precision of immunotherapy and enabling more personalized cancer management. Here, we present a Comprehensive Oncological Biomarker Framework that integrates genetic and molecular testing, imaging, histopathology, multi-omics, and liquid biopsy to generate a molecular fingerprint for each patient. This holistic approach supports individualized diagnosis, prognosis, treatment selection, and response monitoring. Incorporating emerging biomarkers, such as microbiome signatures, further refines patient stratification, guiding the optimization of therapy. By uniting molecular insights with clinical and social factors, this framework aims to address tumor heterogeneity and immune evasion, ultimately improving patient outcomes through precision oncology.

## 1. Introduction

Cancer remains a leading global health concern. Immunotherapy has transformed treatment paradigms, significantly improving survival rates [1]. Immunotherapy, which enhances the body’s natural defenses to recognize and eliminate malignant cells, has revolutionized cancer treatment and revitalized the field of tumor immunology [2]. It represents a paradigm shift by focusing on the “biologic passport” of individual tumors rather than their site of origin [3]. Various immunotherapy approaches, including immune checkpoint inhibitors (ICIs), cancer vaccines, immunomodulators, monoclonal antibodies, oncolytic viral therapy, and cellular therapies such as natural killer (NK) cell therapy and chimeric antigen receptor T-cell (CAR-T) therapy, are employed [2,4] (Figure 1).

Over the last decade, immunotherapy has revolutionized cancer care, resulting in improved survival rates and enhanced quality of life for many patients [5,6]. For example, ICIs have been highly effective in treating several types of cancer [7,8]. As a result of expanded FDA approvals, the proportion of U.S. cancer patients eligible for ICI therapy increased dramatically from 1.5% in 2011 to 43.6% in 2018 [9]. These agents have improved progression-free survival (PFS), overall response rates (ORR), and overall survival (OS) in cancers such as urothelial carcinoma, colorectal cancer, and esophageal cancer [10,11,12,13,14,15,16,17,18,19]. However, many tumors exhibit primary resistance or develop acquired resistance to ICIs, necessitating the development of novel patient stratification strategies based on predictive biomarkers. However, several solid tumors are refractory to ICIs due to complexities in their tumor microenvironment (TME) composition, physical properties and differential immune infiltration [20,21,22]. An additional caveat is the lack of biomarkers in the TME that can predict response rates of ICIs.

Currently, biomarkers such as programmed death-ligand 1 (PD-L1) expression, microsatellite instability (MSI), and tumor mutational burden (TMB) are used to guide immunotherapy, but their predictive accuracy is limited. Ongoing research is focused on developing and validating new biomarkers, including gene expression profiles, immune cell composition, blood-based signatures, and gut microbiome profiles, to improve patient stratification and treatment precision [23].

While immunotherapy can activate the immune system against cancer, it may also cause immune-related adverse events (irAEs) (summarized in Table 1), which affect about 20% of patients and can impact multiple organ systems [6,24,25,26]. These range from mild dermatologic and gastrointestinal symptoms to severe myocarditis, thyroiditis, interstitial lung disease, and psychiatric or rheumatic conditions [6,24,27,28,29,30]. Hyperprogression, or accelerated tumor growth following ICI treatment, is another serious concern. Effective management of these toxicities is crucial for optimizing treatment outcomes and preserving quality of life. Current guidelines from the American Society of Clinical Oncology (ASCO) highlight the urgent need for biomarkers that predict both therapeutic efficacy and adverse effects, thereby enabling truly personalized treatment regimens [31,32,33]. Effectively managing ICI-related toxicities is crucial for optimizing treatment outcomes and preserving patient quality of life.

Recent advances in oncology have catalyzed the development of integrative strategies for incorporating biomarkers into clinical decision-making [34,35,36,37,38]. In this review, we propose a comprehensive framework that unifies diverse biomarker categories—including molecular signatures and gut microbiome profiles—to enhance patient stratification and inform therapeutic decision-making. Moreover, by encompassing molecular and genetic profiling, imaging, histopathology, immunohistochemistry, proteomics, metabolomics, lipidomics and liquid biopsy, this framework collectively provides clinicians with robust and actionable tools to guide clinical trial design, refine patient selection, and personalize treatment regimens (Figure 1). Thereby, it maximizes therapeutic efficacy and safety while simultaneously addressing the inherent limitations of single-biomarker approaches.

## 2. Biomarkers Detection

A variety of techniques are used to detect biomarkers, including proteins, nucleic acids, and metabolites from both liquid and tissue samples. These technologies advance the continuum from biomarker discovery to clinical validation, addressing key challenges and enabling innovation. Improvements in specificity, sensitivity, and reproducibility are expanding the clinical impact of biomarker research.

### 2.1. Immunohistochemistry (IHC) and In Situ Hybridization (ISH)

IHC and ISH are foundational methods for visualizing molecular targets within tissues and are widely used in both research and clinical settings. While these techniques provide precise localization of biomarkers, discrepancies between them remain, driving ongoing efforts to improve their reliability and accuracy [39,40].

### 2.2. Biosensors

Biosensors, particularly immunosensors, provide high sensitivity, rapid detection, and non-invasive biomarker analysis. Using biorecognition elements and signal transducers, these sensors convert biological events into measurable electrical signals [41]. They are categorized into immunosensors, genobiosensors, aptasensors, and enzymatic biosensors. These sensors convert biological events into measurable electrical signals. Biosensors are classified into immunosensors, genosensors, aptasensors, and enzymatic biosensors, enabling the detection of enzymes, antibodies, peptides, aptamers, and microRNAs [42]. However, contamination and non-specific adsorption can cause false positives. Advances in nanomaterials and microfluidics have enhanced biosensor sensitivity, selectivity, and clinical applicability [43,44,45,46,47,48].

### 2.3. Enzyme-Linked Immunosorbent Assay

Enzyme-linked immunosorbent assay (ELISA) is a cornerstone of biomarker detection, relying on the immobilization of antibodies or antigens on solid surfaces for the quantification of antigens or antibodies [49,50]. Despite its widespread use, ELISA faces challenges such as reduced protein activity, non-specific interactions, and the potential for cross-reactivity, which can lead to false positives or negatives [51,52]. Innovations such as streptavidin-biotin complexes and smaller molecule labeling enhance sensitivity and specificity [53]. Proper methodological management is crucial for maintaining the reliability of ELISA in clinical settings [54,55]. The mass-sensing BioCD protein array addresses ELISA’s variability issues, utilizing a SiO_2_/Si wafer to convert protein mass into reflectance variations for concentration measurements [56,57]. While effective, this method has low specificity in complex samples and struggles with detecting low-abundance biomarkers due to masking by abundant proteins [56,57]. These limitations, along with high costs, underscore the need for improvements in specificity and sensitivity [58,59].

### 2.4. Surface-Enhanced Raman Spectroscopy (SERS)

SERS leverages both electromagnetic and chemical enhancements at metal surfaces for ultra-sensitive biomarker detection in complex biological samples [60]. It distinguishes structurally similar molecules, which are critical for the detection of specific cancer biomarkers [61,62]. Gold and silver nanoparticles (AuNPs and AgNPs) are frequently used as enhancing agents, with hybrid designs, such as Au-Ag core–shell nanoparticles, addressing aggregation issues [63,64]. Stability challenges in complex biological environments have been mitigated by encapsulating nanoparticles with polyethylene glycol (PEG) layers, improving performance under varying conditions [65,66]. SERS has been widely utilized for biomarker detection due to its exceptional sensitivity, low sample requirements, robust multi-target detection capability, and well-regulated background interference [67,68,69]. However, SERS faces issues with substrate stability and reproducibility [70].

### 2.5. ATLAS-seq Technology

ATLAS-seq (Aptamer-based T Lymphocyte Activity Screening and SEQuencing) represents a novel approach that combines single-cell technology with aptamer-based fluorescent molecular sensors to identify antigen-reactive T cells. The technology enables more effective identification of TCRs with high functional activity for cancer immunotherapy [71].

## 3. Types of Biomarkers in Cancer Immunotherapy

Biomarkers are defined as measurable characteristics that indicate normal biological processes, disease states, or responses to therapeutic interventions [72]. They can be derived from radiographic, physiologic, histologic, or molecular features [73]. Among these, circulating biomarkers—such as circulating tumor cells, cell-free DNA (cfDNA), RNA, and extracellular vesicles (EVs)—offer valuable insights into tumorigenesis and metastasis, providing a real-time window into tumor dynamics during treatment and disease progression [74,75,76].

In cancer immunotherapy, biomarkers are essential for tailoring treatments and predicting patient outcomes. As summarized in Figure 1, biomarkers are commonly classified into five main categories: diagnostic, screening/susceptibility, predictive, pharmacodynamic, and preventive biomarkers. Ongoing research has focused on developing specific and sensitive biomarkers for oncology, leveraging advanced analytical biochemistry. Innovations in early-stage detection technologies have contributed to reductions in cancer-specific mortality rates. By improving the precision and effectiveness of immunotherapy, biomarkers play a critical role in enhancing patient outcomes [77]. By enhancing the precision and effectiveness of immunotherapy, biomarkers play a crucial role in improving patient outcomes (Figure 1).

Our proposed comprehensive oncological framework emphasizes the integration of diverse biomarkers into a holistic approach, equipping clinicians with tools to address each patient’s unique needs. The integration of multi-omics approaches and computational models further refines biomarker-based predictions, supporting more accurate patient stratification and treatment strategies. This strategy enhances the precision of treatment selection, optimizes response assessment, and helps mitigate the risks of severe adverse events, ultimately enabling truly personalized care.

### 3.1. Diagnostic Biomarkers

Diagnostic biomarkers are molecules that indicate the presence of cancer or disease, guiding diagnosis, monitoring, and treatment selection. These markers include tumor-specific and circulating biomarkers, as well as monitoring and treatment selection. These markers include tumor-specific and circulating biomarkers. For instance, Programmed Death-Ligand 1 (PD-L1) expression is the only FDA-approved biomarker for predicting treatments targeting the Programmed Death-1 (PD-1)/PD-L1 pathways, aiding in therapeutic decision-making [78]. Drugs such as pembrolizumab, nivolumab, and atezolizumab have approvals based on PD-L1 expression levels for various types of cancer. Pembrolizumab is a first-line treatment for Non-Small-Cell Lung Cancer (NSCLC) with PD-L1 expression of 50% or higher and is approved for gastric cancer with a Combined Positive Score (CPS) of 1 or higher. In contrast, nivolumab is used across multiple cancers, sometimes requiring PD-L1 testing (e.g., CPS ≥ 5 in gastric cancer). Atezolizumab is approved for PD-L1-positive triple-negative breast cancer in combination with nab-paclitaxel [79,80,81]. This highlights an urgent need for more precise biomarkers in immuno-oncology [82,83]. In the category of circulating diagnostic markers, a diverse range of molecules has been utilized, including proteins such as CEA, carbohydrates like CA 19-9, nucleic acids like circulating RNAs, circulating tumor cells, and extracellular vesicles [84,85,86,87,88]. These markers have been employed for a long time, and numerous studies continue to explore their potential.

Applying diagnostic biomarkers in clinical decision-making requires careful consideration to strike a balance between accuracy and patient outcomes. For rare cancers, such as ovarian or pancreatic cancer, diagnostic biomarkers must exhibit low false-positive rates to minimize unnecessary psychological distress and invasive procedures. In contrast, for more prevalent cancers like breast or prostate cancer, biomarkers should prioritize low false-negative rates to ensure early detection and timely intervention, enhancing the chances of successful treatment [73].

Although single markers are well-established and widely used as diagnostic tools, combined biomarker panels have demonstrated superior performance [89]. For instance, the combined detection of AFP with cfDNA enhances the specificity of HCC diagnosis, surpassing AFP alone by offering higher sensitivity and better clinical correlation [90]. In addition, detection of serum agrin in combination with secreted IL6 also served as a better prognostic marker for HCC [91]. The advantages of biomarker panels are particularly evident when they reflect changes in independent pathways. For example, combining periostin (POSTN) with CA15-3 and CEA significantly improves the diagnostic performance for breast cancer compared to using CA15-3 and CEA alone [92]. Within the comprehensive oncological framework, diagnostic biomarkers play a pivotal role not only in confirming cancer presence but also in predicting patient responses to treatments such as ICIs, as outlined in Table 2.

### 3.2. Screening/Susceptibility Risk Markers

Susceptibility or risk biomarkers indicate an individual’s likelihood of developing cancer, even in the absence of a current diagnosis. These markers enable both personalized risk assessment and population-level stratification, supporting early detection and timely intervention [93]. For example, germline mutations in tumor suppressor genes such as Breast Cancer Gene 1 and 2 (BRCA1 and BRCA2) are well-established biomarkers for breast and ovarian cancers, guiding personalized screening strategies. Women with these mutations are often advised to undergo earlier and more frequent screening than the general population to improve outcomes [94].

Traditional risk factors have long been used to identify individuals at higher cancer risk. However, advances in genomics, proteomics, lipidomics and microbiomics have led to the development of emerging biomarkers that can significantly improve the sensitivity and specificity of screening panels. For instance, human leukocyte antigen (HLA) class I genotypes influence immunotherapy responses, highlighting the importance of genetic background in cancer risk and treatment outcomes [95]. Additionally, specific T cell states associated with checkpoint immunotherapy response in melanoma provide deeper insight into the cellular mechanisms driving efficacy, paving the way for more personalized immunotherapy strategies. Gut-microbiome-derived biomarker panels have also shown promise in enhancing colorectal cancer (CRC) screening when combined with existing tests such as fecal immunochemical tests (FITs) and guaiac-based fecal occult blood tests (gFOBTs) [96].

In immunotherapy, genetic variations have been linked to both treatment efficacy and immune-related adverse events (irAEs). For example, duplications and deletions in genes such as SMAD3, JAK2, PRDM1, CTLA4, and PDCD1 have been associated with immune-related adverse events (irAEs) in melanoma, underscoring the importance of genetic background in predicting outcomes and managing risks. These findings support the development of more personalized immunotherapy strategies based on an individual’s genetic and immunological profile [97].

Liquid biopsy technologies, such as circulating tumor DNA (ctDNA), provide a minimally invasive approach for detecting and monitoring malignancies. By providing real-time insights into tumor dynamics and the molecular landscape, ctDNA and other circulating biomarkers can refine screening paradigms, enable earlier detection, and allow for better risk stratification of patients [98]. Integrating susceptibility and screening markers into a comprehensive oncological framework will help clinicians identify high-risk individuals who may benefit from early interventions, including immunotherapy [99].

### 3.3. Predictive Markers

Predictive biomarkers in oncology provide information about a patient’s likelihood of responding to specific therapy, guiding treatment decisions and helping to anticipate potential adverse reactions. These markers play a central role in optimizing clinical outcomes by enabling more precise patient selection [100,101,102]. Biomarkers such as microsatellite instability-high (MSI-H), mismatch repair deficiency (dMMR), high tumor mutational burden (TMB ≥ 10 mutations/megabase), and PD-L1 expression are approved by the FDA for clinical use [103,104,105,106]. A recent systematic review and meta-analysis highlight that PD-L1 expression is linked to improved ORR, overall survival (OS), and PFS in patients with metastatic urothelial carcinoma mUC treated with ICIs [107]. However, the study concludes that PD-L1 expression is unlikely to serve as a reliable predictive biomarker [107], underscoring the urgent need for more robust predictive markers to identify patients most likely to benefit from ICIs while minimizing toxicity and financial burden.

Combining multiple biomarkers—such as TMB, PD-L1 expression, and microbiome signatures—improves prediction accuracy [79,108,109,110,111,112]. Predictive biomarkers, including MSI-H and dMMR, indicate impaired DNA repair mechanisms and increased neoantigen production, as demonstrated by pembrolizumab’s 34.3% response rate in the KEYNOTE-158 study and the durable responses observed in colorectal cancer in the CheckMate-142 trial [113,114]. High TMB has also been associated with better outcomes in KEYNOTE-158 [16,115]. Additionally, predictive models that incorporate PD-L1, TMB, and neutrophil counts have demonstrated improved efficacy for ICIs in non-small-cell lung cancer (NSCLC) [116]. 

Combination therapies, such as nivolumab and ipilimumab, emphasize the need for biomarkers to mitigate severe adverse events [103,117]. Emerging studies highlight the gut microbiome as a predictive biomarker, with species like *Akkermansia* enhancing ICI efficacy and reducing toxicity, while others like *Escherichia coli* predict poorer outcomes [118,119,120].

Despite advances, predictive biomarkers face challenges due to population variability, TME dynamics, and tumor heterogeneity. Understanding resistance mechanisms, including mutations in the interferon gamma (IFN-γ) signaling pathway and the PI3K-AKT-mTOR pathway, is essential for improving outcomes [121,122,123,124,125,126,127,128,129]. Serum tumor markers and liquid biopsy approaches also show promise for monitoring the progression of advanced cancer treatment. Technological innovations, such as ATLAS-seq, now enable more precise identification of antigen-reactive T cell receptors (TCRs), supporting a more tailored assessment of immune responses.

Serum tumor markers and liquid biopsies have shown promise in monitoring the treatment of advanced lung cancer [130]. Furthermore, technological advancements like ATLAS-seq have significantly enhanced the identification of antigen-reactive TCRs, providing a more precise assessment of immune responses against cancer cells. This microfluidic single-cell approach enables the isolation of individual T cells that specifically react to tumor antigens, offering new avenues for refining immunotherapy strategies.

A holistic approach that integrates genetic, molecular, epigenetic, and immune-related biomarkers—potentially analyzed with artificial intelligence and machine learning—will likely provide the most accurate predictions of immunotherapy response. Future research should focus on validating multidimensional biomarker panels across diverse patient populations to fully realize the potential of precision oncology (Figure 2).

### 3.4. Pharmacodynamic Biomarkers

Pharmacodynamic biomarkers are molecular indicators that provide real-time information on how a drug interacts with its target and elicits a biological response. Unlike predictive biomarkers, which help identify suitable candidates for immunotherapy, pharmacodynamic biomarkers offer insights into the biological activity and efficacy of therapeutic agents during treatment (see Table 3). These biomarkers are crucial for monitoring treatment effectiveness and enabling timely adjustments to therapeutic strategies, thereby improving patient outcomes.

Pharmacodynamic biomarkers provide valuable insights into the mechanisms of action and target effects of drugs. For example, Liao et al. demonstrated that 18F-fluoroestradiol levels measured by positron emission tomography (PET) can serve as pharmacodynamic biomarkers to assess the response to hormonal therapy in metastatic or recurrent estrogen-positive breast cancer [131]. Similarly, phosphorylated protein kinase B (PKB) has been used to evaluate PI3K pathway inhibition in response to anti-cancer PI3K inhibitors for solid tumors and lymphoma [132]. In immunotherapy, Lambert et al. conducted a phase I trial of Budigalimab (an anti-PD-1 agent) in head and neck squamous cell carcinoma (HNSCC) or NSCLC, finding that early changes in soluble biomarkers such as IFN-γ and CXCL9 within the first 24 h correlated positively with progression-free survival (PFS), while elevated IL-8 levels showed a negative correlation [133]. These findings highlight the potential of pharmacodynamic biomarkers to inform adaptive dosing and enhance therapeutic efficacy. Despite their promise, pharmacodynamic biomarkers remain underutilized in clinical trials. Among 386 phase I immuno-oncology trials, 26% reported no pharmacodynamic biomarker assessments. Even when such biomarkers were evaluated, their correlation with clinical activity was rarely cited in subsequent trials. This indicates limited integration of pharmacodynamic findings into later stages of drug development [134].

### 3.5. Preventive Biomarkers

The identification of cancer preventive biomarkers is an emerging focus in oncology, supporting strategies to reduce cancer incidence. Short-chain fatty acids (SCFAs), mainly acetate, propionate, and butyrate produced by gut microbiota, are promising candidates [135,136]. Mechanistically, SCFAs exert their effects through several pathways. Butyrate functions as a histone deacetylase (HDAC) inhibitor, leading to hyperacetylation of histones and re-expression of tumor suppressor genes such as p21 and p53, which promote cell-cycle arrest and apoptosis [137,138]. SCFAs also engage G-protein–coupled receptors inhibit cell proliferation, induce apoptosis, and cycle arrest via the NF-κB, MAPK, ERK1/2, PI3K, and Wnt signaling pathways [139,140]. Moreover, they can improve therapeutic efficacy of chemotherapy by enhancing the sensitivity of cancer cells to ferroptosis [141,142]. In colorectal cancer (CRC), butyrate provides an energy source for normal colonocytes but induces apoptosis in cancerous colonocytes, exploiting the “butyrate paradox” [143,144]. These attributes position SCFAs as potential biomarkers and therapeutic adjuncts in cancer prevention and treatment.

Despite their promise, preventive biomarkers are still underdeveloped in oncology. Incorporating them into a comprehensive oncological framework will be crucial for advancing early intervention strategies and improving long-term cancer prevention efforts.

## 4. Advancing Cancer Immunotherapy Through Companion Diagnostics

Companion diagnostics are essential for identifying patients most likely to benefit from specific immunotherapies, thereby improving treatment precision and outcomes. The FDA has approved eleven immune checkpoint inhibitors (ICIs) for cancer therapy, including pembrolizumab, nivolumab, cemiplimab, dostarlimab, and retifanlimab (targeting PD-1); atezolizumab, durvalumab, and avelumab (targeting PD-L1); ipilimumab and tremelimumab (targeting CTLA-4); and relatlimab (targeting LAG-3) [145,146,147,148]. For example, PD-L1 expression testing guides the use of pembrolizumab in non-small-cell lung cancer (NSCLC), significantly improving patient outcomes [149]. In small cell lung cancer (SCLC), biomarkers such as NSE and DLL3 have emerged as targets for antibody-drug conjugates and immune-based therapies like Tarlatamab [150,151]. In Hodgkin lymphoma, the expression of CD30 and PD-1 has enabled successful immune checkpoint blockade, whereas in non-Hodgkin lymphoma, the mutations of CD20 and BCL2 inform antibody-based and BCL2 inhibitor therapies [152,153]. Gastrointestinal cancers also benefit from biomarker-driven immunotherapy; for example, HER2 overexpression and PD-L1 expression in stomach and esophageal cancers have facilitated the adoption of monoclonal antibody therapies and ICIs [154,155]. In triple-negative breast cancer (TNBC), BRCA mutations and PD-L1 expression have led to the development of targeted antibody and immune-based treatments [156].

Ongoing research continues to identify novel biomarkers that enhance the precision of immunotherapy. Recent studies show that circulating CD4+ T-cell subsets correlate with progression-free and overall survival in NSCLC patients receiving anti-PD-1 therapy, underscoring the value of immune cell profiling as a predictive tool [157]. The continued identification and validation of such biomarkers are driving innovation in cancer immunotherapy, enabling more tailored and effective treatments.

## 5. Gut Microbiota: A Paradigm Shift in Oncology

Gut microbiota research has broadened the focus of oncology from tumor cells and the host alone to the complex ecosystem of microorganisms within the human body. This expanded perspective deepens our understanding of cancer biology and opens new avenues for developing innovative therapies, preventive strategies, and diagnostic techniques. Conventional oncology biomarkers often emphasize tumor-specific features or host immune responses, overlooking the crucial influence of gut microbiota on these interactions. Gut microbiota contributes to immune system development from birth, shaping immune responses through early microbial exposure and maturation. They can regulate the proliferation and expression of immune cells, particularly the balance between T helper 17 (Th17) and regulatory T (Treg) cells [158]. During immune checkpoint inhibitor (ICI) therapy, specific bacterial species, such as Akkermansia, and microbial metabolites may enhance treatment efficacy by modulating immune cell function and promoting antitumor immunity [159,160,161] (Figure 3). Integrating microbiome analysis into cancer care represents a major advance in precision medicine, offering comprehensive insights into treatment outcomes and reducing therapy-related toxicities. Gut microbiota signatures, such as specific bacterial metabolites, are promising non-invasive diagnostic tools for assessing cancer risk and predicting treatment responses. While fecal microbiota biomarkers have proven useful in colorectal cancer screening, their broader applications in other cancers require further investigation. The gut microbiome also enhances established screening methods, such as fecal immunochemical tests (FITs) and stool DNA (sDNA) testing, by improving sensitivity and specificity in colorectal cancer detection. Ongoing research into microbial influences on cancer recurrence and progression may yield improved strategies for real-time disease monitoring and personalized management [162,163].

### 5.1. The Microbiome: Beyond Bacteria

Microbiome research now recognizes that viruses, fungi, and protozoa—beyond bacteria—play important roles in cancer biology and therapy [164,165]. For example, fungal signatures across cancer types (e.g., Cladosporium enriched in breast cancer; Aspergillus in lung cancer), suggesting prognostic value [166]. Moreover, fungi interact with bacteria and modulate host immune responses [167]. Viruses such as EBV, HPV, HBV, HCV, and HTLV-1 contribute to cancer through direct oncogenic effects or chronic inflammation, while the broader virome may indirectly influence carcinogenesis by modulating immune development and interacting with bacterial communities [168]. Moreover, the interplay among fungi, viruses, and bacteria plays a critical role in shaping host immune responses and cancer outcomes [167,169]. Incorporating non-bacterial microbiome components into cancer immunotherapy could reveal new therapeutic targets to enhance treatment efficacy and overcome resistance [170,171,172,173]. Thus, expanding microbiome research beyond bacteria is essential for a more comprehensive understanding of tumor biology and for developing next-generation cancer diagnostics and therapies.

### 5.2. Microbiota-Driven Immune Modulation

The gut microbiota plays a crucial role in regulating the immune system, influencing both diagnostic and therapeutic outcomes in cancer. It influences immune responses by regulating T-regulatory cells, suppressing myeloid-derived suppressor cells (MDSCs), and activating adaptive immunity through toll-like receptor signaling. By modulating systemic inflammation and cytokine production, the gut microbiota has a significant impact on cancer progression and treatment effectiveness [174,175,176]. Dysbiosis—an imbalance in gut microbiota—has been linked to neuroinflammation, altered neurotransmitter production, and overactivation of the hypothalamic–pituitary–adrenal (HPA) axis, contributing to mood disorders such as depression [177]. The gut–brain axis also influences cancer outcomes through mechanisms such as SCFA production, modulation of immune pathways, and activation of the vagus nerve [177].

A diverse gut microbiome enriched with beneficial bacteria, such as Ruminococcaceae and Faecalibacterium, is associated with improved responses to immune checkpoint inhibitors (ICIs). In contrast, reduced microbial diversity and an overabundance of harmful bacteria, such as Bacteroidales, are associated with the diminished efficacy of immunotherapies. Antibiotic-induced dysbiosis further disrupts microbial balance and reduces the effectiveness of ICI, highlighting the importance of maintaining a healthy microbiome during cancer treatment [5,178,179,180,181,182,183,184,185].

### 5.3. Microbiome Modulation as a Therapeutic Strategy

Manipulating microbiomes is an emerging strategy for improving cancer treatment outcomes and reducing side effects. Approaches include dietary modifications, prebiotics, probiotics, and fecal microbiota transplantation (FMT). Probiotic strains such as Lactobacillus and Bifidobacterium have shown potential to boost immune activation and enhance ICI effectiveness in preclinical models. However, excessive or inappropriate use of probiotics may disrupt microbial balance and compromise treatment efficacy [186,187,188]. Prebiotics, including dietary fibers such as inulin, selectively stimulate the growth of beneficial bacteria, thereby supporting an immune-favorable environment [189,190].

FMT has gained attention for its ability to restore microbial diversity and improve ICI efficacy in patients resistant to standard treatments. However, FMT poses challenges, including risks of pathogen transmission and variability in donor microbiota, necessitating stringent screening and standardized protocols [191,192,193,194,195,196,197]. Preclinical and clinical evidence underscores the importance of microbiome modulation in enhancing anti-tumor immunity and maximizing the effectiveness of immunotherapy [198,199,200,201].

Beyond traditional microbiome-targeting strategies, advances in multi-strain bacterial therapies, developed through comparative genomics and systems biology, provide a more refined and personalized approach. These therapies leverage the functional roles of microbial ecosystems across individuals to create targeted therapeutic candidates [171,193,202,203]. Advances in bacterial genetic engineering enable the creation of customized microbes that can deliver immune-stimulating molecules or modulate metabolic pathways, offering a more precise alternative to FMT for enhancing cancer therapy [204,205,206,207].

A growing understanding of how microbial species and their metabolites influence immune responses is opening new avenues for innovative cancer therapies. The interplay between the gut microbiome and the immune system is now recognized as a key driver of systemic immune regulation, offering opportunities to refine cancer immunotherapy strategies [208]. Recent research highlights the gut microbiome’s ability to enhance treatment efficacy; interventions such as probiotics and FMT have shown promise in increasing the diversity and abundance of beneficial gut bacteria. These approaches have been linked to improved treatment responses and reduced toxicity in cancer patients [209,210].

To fully harness the therapeutic potential of microbiome modulation, a comprehensive oncological framework is essential—one that systematically incorporates microbiome profiling, host–microbe interactions, and metabolomic signatures into clinical decision-making. Such a framework would provide critical insights into how individual microbial compositions influence treatment outcomes, allowing for more personalized and predictive cancer therapies.

### 5.4. Biomarker Bank: Unveiling the Interplay of Cancer Immunotherapy and Gut Microbiome

Biobanks are critical for advancing research on gut microbiome and cancer immunotherapy by providing well-organized clinical and microbiome data that link microbial compositions to treatment outcomes. Modern biobanks have evolved from project-specific repositories into comprehensive infrastructures that integrate biological samples with detailed clinical data, enabling large-scale validation studies to identify reliable biomarkers across diverse populations [211]. Despite their potential, biobanks face challenges such as inconsistent sample standardization, complex data interpretation, regional regulatory differences, and ethical concerns regarding data privacy and usage. Addressing these challenges requires the development of standardized protocols, the adoption of consistent bioinformatics tools, and the establishment of sustainable funding through collaborations with stakeholders to ensure the long-term viability of biobanks [212,213,214]. By overcoming these barriers with standardized methodologies, advanced data analysis techniques, sustainable funding models, and strict ethical guidelines, biobanks can deepen our understanding of how the gut microbiome influences cancer immunotherapy. Ultimately, this will pave the way for more personalized and effective cancer treatments.

### 5.5. Integrating Gut Microbiota into Comprehensive Oncological Frameworks

Integrating insights into gut microbiota into oncology requires a holistic approach that views the human body as a complex ecosystem where microorganisms interact with host processes. To realize the therapeutic potential of the microbiome, challenges such as individual variability in microbiome composition, temporal fluctuations, and methodological inconsistencies must be addressed. Collaborative efforts are needed to standardize assessments and conduct large-scale studies that reveal clinically relevant microbiome signatures. This paradigm shift has the potential to revolutionize cancer care by improving treatment accuracy, enabling early preventive interventions, and advancing diagnostic techniques.

Microbiome-driven strategies in oncology can transform patient outcomes by tailoring treatments based on microbial profiles, moving the field toward truly personalized cancer care. A comprehensive oncological framework should emphasize the gut microbiome’s role in regulating immune responses, enhancing antitumor immunity, and improving the efficacy of cancer immunotherapy. Continued research and innovation, including multi-omics integration and computational modeling, will be essential to unlock the full therapeutic potential of the gut microbiota, ultimately leading to improved survival rates and enhanced treatment efficacy for cancer patients.

## 6. Liquid Biopsy: Promises and Hurdles

A liquid biopsy is a minimally invasive technique that utilizes a simple blood sample, thereby reducing patient discomfort and the risks associated with surgical tissue extraction. Its non-invasive nature allows for frequent sampling, enabling clinicians to monitor disease progression and treatment response in real time. By capturing genetic information from multiple tumor sites, liquid biopsy provides a more accurate representation of tumor heterogeneity than single-site tissue biopsies. Additionally, liquid biopsies offer faster turnaround times, supporting timely clinical decision-making and potentially improving patient outcomes [215].

Circulating tumor DNA (ctDNA) analysis represents a significant technological advancement in liquid biopsy, offering valuable insights into cancer type, grade, and progression [216]. Found in bodily fluids such as blood plasma, urine, and cerebrospinal fluid, ctDNA is derived from apoptotic or necrotic cancer cells. This biomarker provides valuable insights into cancer type, grade, and progression. ctDNA may help select patients who will benefit from adjuvant chemotherapy, and multiple clinical trials are actively underway [217]. ctDNA-based liquid biopsies provide a comprehensive view by capturing genetic information from both primary tumors and metastatic sites, making them an effective non-invasive tool for disease assessment. Additionally, the ability to collect multiple samples over time makes liquid biopsy an ideal tool for monitoring disease progression dynamically [218,219].

Extracellular vesicles (EVs), nano-sized particles released by cells, have emerged as novel tools in cancer diagnostics due to their ability to transfer biomolecules, including proteins, lipids, and nucleic acids, between cells. EVs are nano-sized particles released by cells that play a crucial role in intercellular communication by transferring biomolecules such as proteins, lipids, and nucleic acids. Previously regarded as cellular waste, EVs are now recognized as key mediators of intercellular communication, playing critical roles in cancer progression by transferring molecular signals between cells [220,221,222]. Recent studies have also identified metabolites and even organelles within EVs, further expanding our understanding of their complex composition and potential functions [223,224]. In cancer, tumor-derived EVs (TEVs) play a pivotal role in disease onset and progression, affecting molecular and cellular pathways, promoting cell migration and invasion, influencing drug responses, and modulating immune functions [221]. Tumor-derived extracellular vesicles (TEVs) play a pivotal role in regulating tumor growth and progression. Their molecular cargo reflects the state of originating cells, making them valuable biomarkers for early cancer detection [225,226]. Circulating tumor cells (CTCs), circulating tumor DNA (ctDNA), and cell-free RNA (cfRNA) are promising diagnostic biomarkers, as illustrated in Figure 4, but they face challenges such as low concentrations in the bloodstream and susceptibility to degradation by nucleases. Additionally, these biomarkers have a short lifespan and are prone to degradation by circulating nucleases [226,227,228].

### Implementation in Precision Medicine

Liquid biopsy has emerged as a valuable tool in precision medicine, demonstrating significant potential across various cancers, including lung, colorectal, breast, and prostate cancer [229]. Despite its promise, integrating liquid biopsy into routine clinical practice faces challenges, including the standardization of protocols, sample handling complexities, and ensuring biomarker stability [230]. Large-scale validation studies are critical for establishing the reliability and accuracy of liquid biopsy methods across diverse patient populations. Establishing standardized protocols is crucial to ensure consistency and clinical relevance [231]. The non-invasive nature of liquid biopsies introduces complexities related to sample collection, handling, and processing. Ensuring biomarker stability and minimizing variability during these stages is vital for accurate interpretation. Integrating liquid biopsy into healthcare systems requires comprehensive training for clinicians, development of standardized guidelines, and robust clinical trial data to validate its impact on patient outcomes [232,233].

Liquid biopsy holds transformative potential by enabling earlier cancer detection and providing real-time insights into disease progression and treatment response (Figure 4). These technologies can significantly improve treatment outcomes by enabling earlier detection and facilitating timely therapeutic interventions, offering a dynamic perspective on disease evolution. Realizing the full potential of liquid biopsy requires collaborative efforts between researchers, clinicians, and regulatory agencies to refine technologies, establish regulatory frameworks, and ensure seamless clinical integration. By addressing current technical and clinical barriers, liquid biopsies can play a central role in personalized cancer prevention strategies that improve early detection and patient care. Future developments will likely focus on increasing sensitivity for detecting low-abundance biomarkers, developing multi-analyte panels that combine different circulating markers, and creating computational approaches that integrate liquid biopsy data with other clinical parameters for more comprehensive patient assessment [232,234].

## 7. Biomarker Integration in Combination Therapy Strategies

Combination therapy employs multiple treatment modalities, such as immunotherapy, chemotherapy, radiotherapy, and targeted agents, to enhance effectiveness by addressing diverse mechanisms of cancer progression [235,236]. This approach is especially important in cancer immunotherapy, where tumor heterogeneity and resistance often limit the success of single-agent therapies. By integrating these diverse treatments, combination strategies aim to improve clinical outcomes, overcome resistance, and expand the range of patients who benefit from advanced therapies [235,236]. Incorporating biomarkers into combination therapy enables more personalized treatment strategies, optimizing therapeutic benefits while minimizing adverse effects [237,238]. Incorporating biomarkers into combination therapy enables the development of personalized treatment strategies that optimize therapeutic benefits while minimizing adverse effects.

Checkpoint-blocking antibodies, including anti-CTLA-4 and anti-PD-1 therapies, have demonstrated promise in overcoming resistance mechanisms in cancer immunotherapy [236]. Dual checkpoint blockade (anti-CTLA-4 plus anti-PD-1) improves response rates compared to monotherapy but is associated with increased side effects and higher relapse risk. Adding immune-enhancing agents, such as cytokines, alongside anti-CTLA-4 antibodies, can yield synergistic benefits, highlighting the potential of multifaceted immunotherapy approaches [193,239,240,241].

The gut microbiome also plays a pivotal role in enhancing the efficacy of anti-PD-1 therapies. Dietary interventions, prebiotics, probiotics, and FMT are under investigation for their potential to boost treatment efficacy. Certain bacterial species, such as Enterococcus hirae and Barnesiella intestinihominis, have been shown to enhance the efficacy of cyclophosphamide by modulating immune responses [242,243,244].

Combining chemoradiotherapy with anti-PD-1 immunotherapy has shown encouraging results in cancers such as NSCLC, gastric cancer, breast cancer, and hematological malignancies. This synergy enhances tumor immunogenicity, facilitates antigen presentation, and promotes the recognition of tumor cells by T cells. Chemoradiation also increases the tumor mutation burden and reshapes the tumor microenvironment to favor antitumor immunity by modulating cellular and molecular components [245,246,247,248,249].

Radiotherapy further modifies gene expression within the tumor microenvironment, increasing mutation burden and promoting antigen presentation to enhance T-cell recognition. In NSCLC, radiotherapy has been shown to upregulate PD-L1 expression, thereby influencing immune responses [250]. However, excessive immune responses have been reported in some cases, particularly in obesity-related cancers like esophageal adenocarcinoma when PD-1 immunotherapy is administered post-radiotherapy [251,252,253,254]. The interaction between radiation and immunotherapy depends on factors such as dose and immune cell dynamics, and further research is needed to optimize protocols and improve outcomes.

### 7.1. Pan-Cancer Approach: Overcoming Challenges of Cancer Heterogeneity and the Tumor Microenvironment

Recent advances in cancer immunotherapy have demonstrated significant potential, yet challenges such as cancer heterogeneity and the complexity of the tumor microenvironment (TME) continue to limit treatment effectiveness. Variations within and between tumors—known as inter- and intra-tumoral heterogeneity—impact genetic, epigenetic, and phenotypic characteristics, leading to diverse immune responses and resistance to therapies (Figure 5). The TME comprises cancer cells, stromal elements such as cancer-associated fibroblasts, immune cells, and the extracellular matrix. These components collectively enable immune evasion and influence treatment outcomes. Thorsson et al. (2018) [255] identified three immune phenotypes—immune-excluded, immune-inflamed, and immune-desert—highlighting the varied interactions between tumors and the immune system. These complexities underscore the need for innovative strategies to develop more universally effective therapies [255]. Together, these complexities highlight the need for innovative approaches to develop more universally effective therapies [255,256].

The pan-cancer approach addresses these challenges by targeting shared molecular and genetic traits across different cancers, aiming to develop therapies with broader applicability (Figure 4 and Figure 5). Instead of focusing on individual cancer types, this strategy identifies common features, such as altered receptor expression (e.g., PD-L1, CTLA-4), metabolic changes (including glutaminase activity), extracellular matrix remodeling (collagen or hyaluronan), and variations in membrane lipid composition. Recent studies from our group have established agrin as a critical ECM biomarker in lung cancer, with implications for prognosis and therapeutic response [91,257,258,259,260,261,262,263,264,265].

These shared traits facilitate the development of pan-immunotherapy strategies, expanding treatment options across diverse tumor types. Innovations such as aptamers or antisense oligonucleotides targeting TME components—for example, atezolizumab combined with VEGF inhibitors in pancreatic ductal adenocarcinoma—are being explored to improve efficacy [256,266,267,268,269]. The identification of several promising pan-cancer biomarkers supports the feasibility of universal therapies across multiple tumor types:Microsatellite instability-high (MSI-H): This is an FDA-approved biomarker associated with a high tumor mutational burden and increased neoantigen production. Pembrolizumab has shown efficacy in MSI-H tumors across various cancers [255].NTRK gene fusions: There are genetic alterations found across diverse cancers. Larotrectinib targets NTRK fusion-positive tumors with demonstrated clinical success in both adult and pediatric patients [256].RET Mutations and Fusions: RET mutations occur in cancers like NSCLC and colorectal cancer. Targeted therapies for RET fusion-positive tumors represent emerging pan-cancer treatment options [269].Tumor Mutational Burden (TMB): It predicts enhanced responses to ICIs like pembrolizumab. Its FDA approval for TMB-high solid tumors underscores its importance as a predictive biomarker [256].PD-L1 Expression: It is a key biomarker for ICI response in melanoma and NSCLC. However, variability in PD-L1 thresholds highlights the need for standardized testing protocols to improve treatment efficacy [255,269].

Refining pan-immunotherapy strategies relies on cutting-edge technologies such as CRISPR-based genome editing, synthetic biology, and artificial intelligence (AI)-driven multi-omics analysis to enhance precision and efficacy. These approaches facilitate the discovery of novel biomarkers and the development of advanced therapeutic strategies, including “living drugs” like CAR-T cells engineered to target shared tumor vulnerabilities. CRISPR technologies enable precise genome alterations, allowing researchers to identify immune evasion mechanisms and design targeted interventions to overcome resistance. Artificial chromosome construction presents a promising avenue for developing personalized therapies by integrating multiple genetic modifications to address shared tumor vulnerabilities [270]. AI-driven analysis enables seamless integration of multi-omics data—including genomics, proteomics, transcriptomics, lipidomics and metabolomics—to identify pan-cancer biomarkers with high precision [271].

Advances in spatial transcriptomics and multiplex immunohistochemistry further complement these technologies by providing insights into spatial interactions within the tumor microenvironment. Leveraging these tools, pan-immunotherapy strategies could lead to the development of more affordable, less toxic treatments that are universally applicable across diverse cancer types.

### 7.2. Challenges and Opportunities in Combination Therapies

Combining immunotherapy with chemotherapy, radiotherapy, or targeted therapy can improve patient outcomes, but these strategies also introduce important challenges that require careful management [272]. A key issue is selecting patients who are suitable for combination therapies to maximize benefits and minimize side effects. Biomarkers are essential for identifying patients most likely to respond, based on the specific characteristics of their tumors and immune systems [273].

Managing the increased risk of side effects, such as immune-related adverse events (irAEs), is another major challenge. Effective monitoring and management protocols are crucial to ensure patient safety. Despite these risks, combination therapies can enhance cancer immunotherapy by targeting multiple pathways, overcoming resistance, activating different components of the immune system, and making tumors more susceptible to immune attack [272,274,275]. To effectively implement such multifaceted treatment strategies, a comprehensive oncological framework is required—one that integrates clinical, molecular, and immunological data to guide decision-making. This framework should provide accessible, real-time information on patient profiles, biomarker status, treatment responses, and potential toxicities, enabling more precise and personalized therapeutic planning.

## 8. Advancing Biomarker Integration in Precision Oncology: Pathways, Challenges, and Innovations

Biomarkers are crucial to advancing personalized and precision medicine by enhancing diagnosis, predicting disease outcomes, and informing treatment decisions. Biomarker development is a stepwise process, beginning with the discovery of significant molecular changes using advanced analytical methods. Promising candidates are then validated across diverse patient groups to ensure clinical relevance and cost-effectiveness. This process enables the creation of diagnostic tools that capture the complex relationship between cancer and the immune system, ultimately supporting better patient selection and treatment planning (Figure 5).

Pathologists play a key role in ordering and interpreting biomarker tests. However, translating biomarkers from research to clinical use is a complex process that requires rigorous scientific, clinical, and regulatory evaluation. Because requirements differ by region, pathologists must possess broad expertise to ensure accurate results and minimize errors [276].

### 8.1. Technological Innovations in Biomarker Research

Recent advances have transformed biomarker research by incorporating technologies such as liquid biopsies, microbiome analysis, and artificial intelligence (AI). For example, in colorectal cancer (CRC), mass spectrometry reanalysis of public proteomics datasets has detailed the proteomic landscape, supporting tumor protein expression assessments and liquid biopsy applications that correlate molecular profiles with patient outcomes [277]. Serum-derived extracellular vesicles (EVs) have enabled the discovery of novel diagnostic biomarkers, as demonstrated by the CRC-EVArray diagnostic model, which provides a scalable solution for CRC screening and clinical translation [278].

Integrating gut microbiota data with AI frameworks further enhances CRC classification, highlighting the microbiome’s influence on human health and paving the way for advanced diagnostics and therapeutics [279]. Combining traditional molecular markers with systems biology approaches, including the gut microbiome, advances biomarker discovery and opens new pathways for innovative cancer prevention strategies. However, translating these findings into clinical practice remains challenging and requires rigorous research and thoughtful application [276].

### 8.2. Addressing Regulatory and Technical Challenges

Inconsistent testing methods and standards present significant regulatory challenges for implementing biomarkers in clinical care. Collaboration with regulatory agencies such as the FDA and EMA is essential to establish unified validation guidelines and standardized protocols [280,281]. Overcoming technical barriers requires developing standardized testing methods, creating centralized biomarker databases, and implementing robust statistical models that account for patient variability. These measures will enhance the reliability and clinical utility of biomarkers [282,283].

### 8.3. Forward Paths and Upcoming Innovations

Cancer immunotherapy research is expanding beyond established biomarkers, such as PD-L1 expression and TMB, to explore neoantigen profiles, circulating tumor DNA, and immune cell infiltration patterns within the tumor microenvironment (Figure 6). Standardizing testing methodologies is crucial for enhancing biomarker reliability across various settings. Initiatives such as the Cancer Immune Monitoring and Analysis Centers and the Cancer Immunologic Data Commons (CIMAC-CIDC) Network exemplify efforts to harmonize protocols across research centers [106,284]. Comprehensive biospecimen collection before and during treatment, coupled with biomarker-based patient stratification, significantly enhances the precision of immunotherapy trials. A deeper understanding of tumor-immune system interactions could identify novel resistance mechanisms and lead to targeted interventions. Advanced technologies—including CRISPR screening, high-throughput sequencing, and multiplex IHC—are expanding the biomarker discovery landscape, while minimally invasive assays like circulating tumor DNA testing offer promising diagnostic and prognostic applications [106,284].

Pharmaceutical companies are advancing precision medicine by analyzing patient samples with next-generation sequencing to integrate multi-omics data and develop microbial modulators. These initiatives, supported by standardized methodologies, are creating pathways to novel therapeutic approaches [285,286,287]. Personalized immunotherapy tailored to specific biomarker profiles is progressing rapidly, with technologies like ATLAS-seq enabling the accurate identification of antigen-reactive TCRs, representing significant advances in precision oncology [71].

Future research must address standardization challenges and establish universal cutoff values for biomarkers to ensure consistent clinical applications [115]. Expanding studies of immune-related adverse events will enhance risk assessment models and patient safety [288,289,290], while non-invasive screening methods such as liquid biopsies and microbiome analysis represent transformative approaches for early detection and monitoring of cancer progression.

### 8.4. The Potential of Comprehensive Biomarker Panels

Combining different types of biomarker data—such as tumor characteristics, immune cell counts, signaling proteins, and patient health status—provides a more comprehensive basis for selecting and tailoring cancer treatments. While biomarkers like PD-L1 expression and tumor mutational burden (TMB) can help predict treatment response, their accuracy is limited by tumor heterogeneity, temporal changes, and inconsistencies in testing methods. Continuous monitoring and advanced standardization are needed to refine their clinical utility [291,292,293,294]. Simple blood tests measuring ratios of immune cells—such as neutrophil-to-lymphocyte or platelet-to-lymphocyte ratios—can help assess patient risk and guide personalized immunotherapy decisions. An integrated cancer care model should unify these diverse biomarkers, leveraging advanced computational tools and machine learning to analyze data and adapt treatment plans as tumors and immune responses evolve.

Developing panels that combine tumor features, immune cell counts, and signaling proteins can help predict both the effectiveness of a treatment and the likelihood of side effects, ultimately making cancer therapy safer and more effective [295] (Figure 7). By overcoming these challenges and adopting new technologies, comprehensive biomarker strategies have the potential to make cancer treatment more precise and beneficial for patients worldwide.

## 9. Conclusions

The Comprehensive Oncological Framework integrates molecular, genetic, and immune system information to enhance diagnosis, guide therapy selection, and monitor treatment response. By accounting for patient variability and tumor diversity, this approach supports the development of new, individualized strategies in clinical practice.

Immunotherapy now plays a central role in cancer treatment alongside surgery, chemotherapy, and radiotherapy. Ongoing research into predictive biomarkers and the gut microbiome underscores the complexity of the immune system’s involvement in cancer progression and therapy response. Factors such as the patient’s immune profile, tumor biology, and disease stage all impact treatment outcomes. Advances in immune checkpoint inhibitors and CAR-T cell therapies have inspired new immunotherapies and strategies to enhance the function of immune cells, including dendritic cells, natural killer cells, and macrophages, thereby improving the targeting of tumors. Researchers are also expanding biomarker discovery to include blood-based and systemic markers, offering a more comprehensive view of treatment effectiveness.

The gut microbiome is an emerging focus in immunotherapy research, though challenges remain in identifying which microbes enhance immune responses and in translating findings from animal models to human patients. Future clinical trials are needed to clarify how microbiome-based interventions can be effectively applied in practice.

Laboratory and clinical studies are actively testing new biomarker-driven approaches across various cancers. By deepening our understanding of immune-tumor interactions, researchers aim to develop more targeted and personalized therapies with fewer side effects. The next phase of cancer immunotherapy will leverage multiple biomarker types, including gut microbiome profiles, to better match patients with optimal treatments. As research advances, microbiome analysis may become a standard component of cancer care, aiding in treatment prediction and reducing side effects. Continued progress in biomarker-driven precision oncology holds the promise of transforming cancer treatment and improving survival for patients.

## 10. Outstanding Questions

Despite significant advances in understanding the interplay between the gut microbiota and immune checkpoint inhibitor (ICI) therapy, several critical questions remain unanswered. Addressing these gaps is essential for translating current findings into more effective and personalized cancer immunotherapies. The following outstanding questions highlight key areas for future investigation:Mechanistic specificity: What molecular pathways enable Akkermansia and microbial metabolites (e.g., inosine, SCFAs) to selectively enhance antitumor CD8+ T cell activity while suppressing Treg-mediated immunosuppression?Biomarker validation: Can standardized microbial signatures (e.g., Akkermansia abundance, metabolite profiles) reliably predict ICI response across diverse cancer types and patient demographics?Therapeutic optimization: How can microbiome-modulating interventions (e.g., probiotics, fecal transplants) be temporally synchronized with ICI administration to maximize efficacy without exacerbating immune-related adverse events?Ecological dynamics: Does sustained ICI efficacy require persistent colonization of therapeutic microbial strains, or can transient microbiome remodeling induce durable immune reprogramming?

## Figures and Tables

**Figure 1 biomolecules-15-01304-f001:**
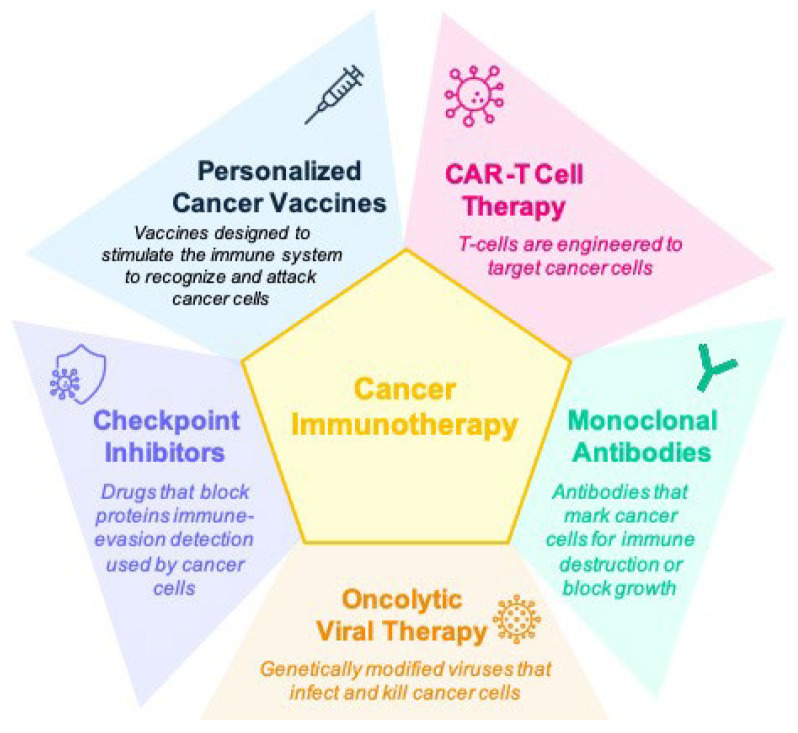
Overview of the five major types of immunotherapies in cancer treatment. This schematic illustrates the principal immunotherapeutic strategies currently applied in oncology. (1) Immune checkpoint inhibitors (ICIs): monoclonal antibodies that block inhibitory receptors to restore T-cell activity against tumor cells. (2) Cancer vaccines: agents designed to elicit or enhance tumor-specific immune responses by presenting tumor-associated antigens. (3) Immunomodulators: cytokines, adjuvants, or agents that broadly stimulate or regulate immune responses within the tumor microenvironment. (4) Monoclonal antibodies (mAbs): laboratory-engineered antibodies that can directly target tumor antigens, induce antibody-dependent cellular cytotoxicity, or deliver cytotoxic payloads. (5) Cell-based therapies: adoptive cell transfer strategies, including natural killer (NK) cell therapy and chimeric antigen receptor T-cell (CAR-T) therapy, which harness and engineer immune cells for enhanced tumor recognition and eradication.

**Figure 2 biomolecules-15-01304-f002:**
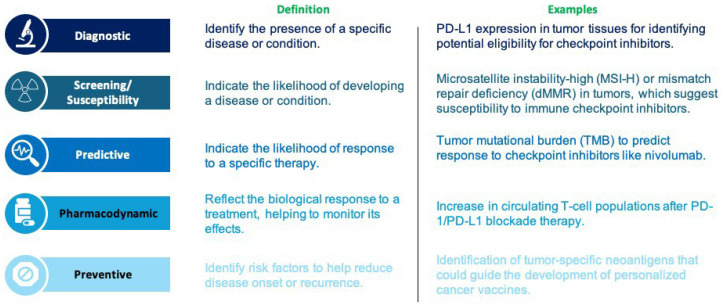
Categories of biomarkers in cancer immunotherapy and representative examples. The main categories of biomarkers related to cancer immunotherapy are compiled in this table, along with their definitions and specific examples. While screening or susceptibility biomarkers, such as MSI-H/dMMR, are used to evaluate the risk of acquiring cancer, diagnostic biomarkers, like PD-L1 expression, are utilized to detect the presence of illness. Therapeutic selection is aided by predictive biomarkers, such as tumor mutational load, which suggest potential therapeutic responses. Preventive biomarkers, such as tumor-specific neoantigens, detect risk factors to guide preventative measures, such as cancer vaccinations, whereas pharmacodynamic biomarkers, such as alterations in circulating T-cell populations, track biological responses to therapy. Together, these indicators allow for a customized approach to cancer immunotherapy.

**Figure 3 biomolecules-15-01304-f003:**
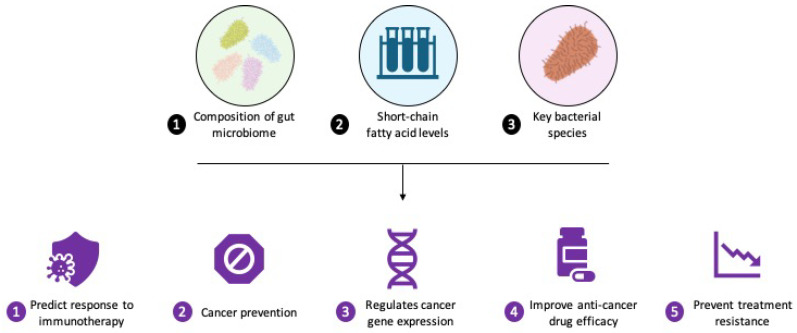
Integrated approach to predictive biomarkers and cancer outcomes. This figure highlights the role of predictive biomarkers, particularly short-chain fatty acids (SCFAs), such as butyrate, in advancing cancer prevention and treatment. SCFAs modulate cancer-related pathways, including gene expression, metabolism, and cellular processes such as autophagy and apoptosis, while also enhancing the efficacy of anti-cancer drugs and reducing treatment resistance.

**Figure 4 biomolecules-15-01304-f004:**
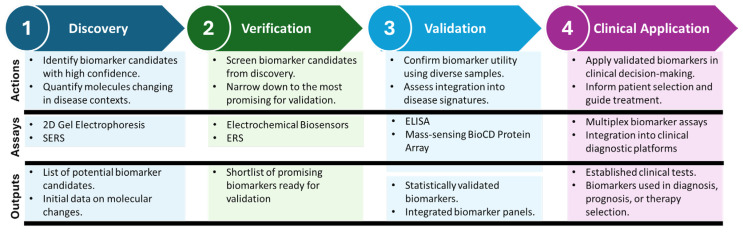
Stages of Biomarker Development: From Discovery to Clinical Application. The process begins with Discovery, where high-confidence biomarker candidates are identified using advanced detection technologies, followed by Verification, which screens and prioritizes potential biomarkers for further investigation. In the Validation phase, biomarkers are rigorously tested using diverse and representative patient cohorts to ensure reliability and predictive power. Finally, biomarkers transition to Clinical Application, where they guide patient selection, inform treatment strategies, and improve clinical outcomes. Each stage highlights specific technologies, such as ELISA, SERS, and electrochemical biosensors, alongside key challenges, including specificity, sensitivity, and reproducibility, emphasizing the iterative and multidisciplinary nature of biomarker development.

**Figure 5 biomolecules-15-01304-f005:**
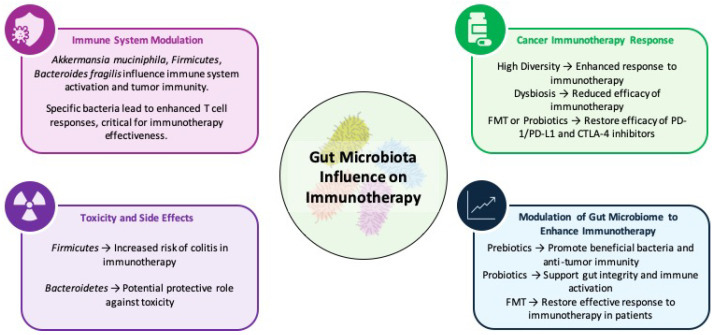
Gut Microbiota Influence on Immunotherapy. The diagram illustrates the dynamic relationship between the gut microbiome and the immune system in the context of cancer immunotherapy. Key microbial taxa, including *Akkermansia muciniphila* and *Bacteroides fragilis*, influence immune responses by promoting T cell activation and modulating the tumor microenvironment (TME). Dysbiosis, or microbial imbalance, is associated with impaired immune responses and reduced efficacy of cancer therapies, such as PD-1/PD-L1 and CTLA-4 inhibitors. Modulation of the gut microbiome through interventions such as fecal microbiota transplantation (FMT) or probiotics has been shown to enhance the effectiveness of immunotherapy. The variability of the gut microbiome across individuals, as well as the challenges in standardizing microbiome assessments, underscores the complexity of using microbiome data as a predictive biomarker for treatment outcomes. Further research and large-scale studies are needed to establish reliable protocols for clinical application.

**Figure 6 biomolecules-15-01304-f006:**
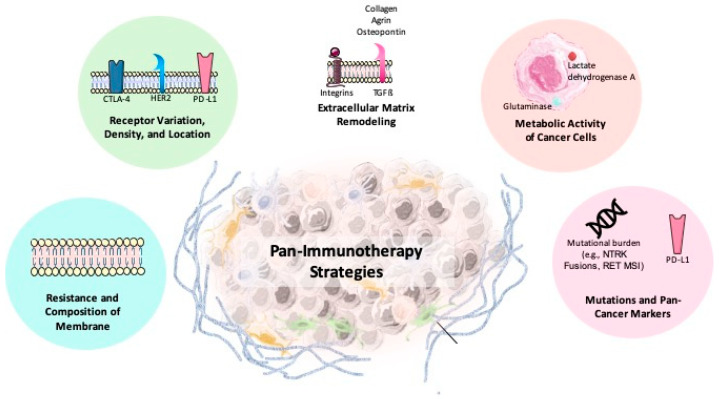
Schematic Overview of Pan-Immunotherapy Strategies Targeting Common Molecular Features of Cancer Cells to Address Tumor Heterogeneity and Overcome Resistance Mechanisms. The tumor microenvironment (TME) comprises several key components, including cancer cells, immune cells, stromal cells, and the extracellular matrix. Common molecular targets across various cancer types are depicted, such as overexpressed receptors (e.g., PD-L1, CTLA-4), altered metabolic pathways, and cancer-specific antigens. The comparison of normal versus cancer cell membrane composition illustrates differences in receptor density, lipid composition, and membrane fluidity. This schematic emphasizes the potential of pan-immunotherapy agents to target multiple molecular features, enhancing the efficacy of immunotherapy across diverse cancer types by exploiting shared vulnerabilities while minimizing toxicity to normal cells. These approaches aim to enhance therapeutic outcomes and expand the applicability of immunotherapy.

**Figure 7 biomolecules-15-01304-f007:**
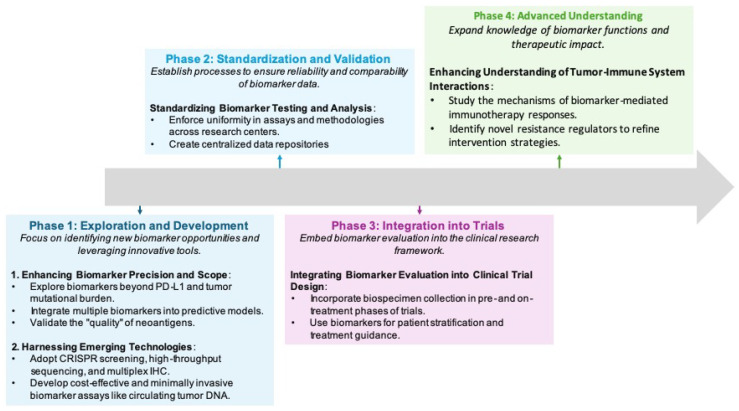
Roadmap for Biomarker Research and Application in Cancer Immunotherapy. The diagram illustrates a phased approach to advancing biomarker discovery, standardization, integration, and understanding in cancer immunotherapy. Phase 1: Exploration and Development focuses on expanding biomarker categories (e.g., neoantigen profiles, circulating tumor DNA) and leveraging emerging technologies (e.g., CRISPR, high-throughput sequencing). Phase 2: Standardization and Validation emphasizes the uniformity of assays and methodologies across research centers, as well as the establishment of centralized data repositories. Phase 3: Integration into Trials highlights the incorporation of biomarker evaluation into clinical trial design, including biospecimen collection and patient stratification. Phase 4: Advanced Understanding delves into the mechanisms through which biomarkers influence immunotherapy responses and the identification of novel resistance regulators. Each phase builds upon the previous one to facilitate precision medicine approaches in cancer immunotherapy.

**Table 1 biomolecules-15-01304-t001:** Classification of possible irAEs and their associated organs.

Organ System	Possible irAEs
Dermatologic	Rash, pruritus, vitiligo, dermatitis
Endocrine	Hypothyroidism, hyperthyroidism, adrenal insufficiency, diabetes
Pulmonary	Pneumonitis, dyspnea, cough
Cardiovascular	Myocarditis, pericarditis, arrhythmias
Hepatic	Hepatitis, elevated liver enzymes
Musculoskeletal	Arthritis, myositis, joint pain
Renal	Nephritis, acute kidney injury
Neurological	Peripheral neuropathy, encephalitis, myasthenia gravis
Hematologic	Anemia, thrombocytopenia, neutropenia

**Table 2 biomolecules-15-01304-t002:** Classification of diagnostic biomarkers for select cancer types.

Cancer Type	Diagnostic Biomarker	Description/Role
Non-Small-Cell Lung Cancer	Gut microbiota profiles (e.g., Bacillus, Bifidobacterium, Faecalibacterium)	Specific gut microbiota compositions can distinguish between NSCLC patients and healthy controls. Diagnostic models using bacterial abundance data have shown high accuracy.
Ovarian, Breast Cancer	CD39 expression	High CD39 expression on tumor-infiltrating lymphocytes (TILs) correlates with an immunosuppressive tumor environment and serves as a diagnostic marker.
Esophageal Squamous Cell Carcinoma	CD39-expressing T cells (combined with a clinical nomogram)	High levels of CD39+ T cells serve as a diagnostic and prognostic biomarker. A panel combining these cells with clinical indicators improves survival and prognosis predictions.

**Table 3 biomolecules-15-01304-t003:** Pharmacodynamic Biomarkers and Their Applications in Cancer Therapy.

Pharmacodynamic Biomarker	Cancer Type	Application
18F-fluoroestradiol (PET imaging)	Metastatic or recurrent breast cancer (ER-positive tumors)	Detects treatment response to hormonal therapy. Enables real-time monitoring of estrogen receptor activity, guiding therapy adjustments in breast cancer.
Phosphorylated AKT	Solid tumors and lymphoma	Measures PI3K signaling inhibition in response to PI3K inhibitor drugs. Reflects the pharmacodynamic effect of targeted therapies, helping in dose optimization for PI3K inhibitors.
IFN-γ and CXCL9	Head and neck squamous cell carcinoma, non-small cell lung cancer	Indicates early immune activation, correlates with increased progression-free survival in Budigalimab therapy. Serve as positive predictive markers in immunotherapy, particularly in PD-1 inhibitors.
IL-8	Head and neck squamous cell carcinoma, non-small cell lung cancer	Negative biomarker for progression-free survival; elevated levels within 24 h correlate with poorer outcomes in Budigalimab therapy. A cautionary biomarker indicating potential treatment resistance or suboptimal response.

## Data Availability

All materials are present in the manuscript text.

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
