# Peer review of "A Comprehensive Oncological Biomarker Framework Guiding Precision Medicine"

_biomolecules, 2025, doi:10.3390/biom15091304_

Round 1
Reviewer 1 Report
Comments and Suggestions for Authors
This manuscript presents an in-depth review of oncological biomarkers across multiple cancer types and proposes a comprehensive framework for integrating these biomarkers into precision medicine strategies. The topic is highly relevant, considering the growing demand for individualized cancer care and biomarker-driven decision-making. The review is extensive and demonstrates a strong effort to cover a broad and complex field. However, the manuscript suffers from issues related to clarity, structure, depth of analysis, and at times, overgeneralization. To improve its quality and impact, it would benefit from clearer organization, and a more focused discussion of translational and clinical implications with some examples or reports.
- In Introduction - Consider adding a sentence clarifying the translational impact: how this framework can directly influence clinical trial or patient management decisions.
- Content of the manuscript is rich but it often reads as a long list of information without thematic coherence or conceptual flow. Also add other reports in each section. Currently it is just write up without proper discussion of some reports e.g. in biomarker detection section - authors can add a table with list of biomarkers for each methods explained. same goes for other section as well.
- Figure 1 legend is not in detail and self-explanatory.
- for each section, information is more kind of general. no specific examples are discussed. e.g in section 3.5 - there is no detailed explanation on possible mechanisms how SCFA production influences cancer progression.
- Section 5.1 - Add some reports of non-bacterial microbiome. consider adding some discussion on the challenges in translating microbiome findings from animal models to human applications - it would add important context.
- In figure 4- there is no write up for no.4. is it clinical applications?
- throughout the manuscript, there are lots of repetitive sentences. Authors must carefully read the whole manuscript and correct this.
- Also, authors must check all the references. e.g in section 8.1 - ref 130 doesn't align with the write up. Authors stated some report for CRC and and cited paper is on lung and doesn't mention what they discussed.
- In general, the review is descriptive in many parts and lacks critical analysis of current challenges, limitations, and controversies. Also, review mentions technical advances but should more clearly contrast their comparative sensitivity/specificity in a clinical context.
- some minor comments - Ensure abbreviations are defined at first mention; include a comprehensive abbreviation list. Maintain consistency in biomarker and technology nomenclature throughout the manuscript.
- All Figures are not so clear. Please submit figures with more resolution and clarity.
The manuscript presents an important and broad topic, but to meet publication standards, it requires structural reorganization, deeper analysis, improved clarity, and stronger visuals. With these revisions, it can serve as a valuable resource for both researchers and clinicians in precision oncology.
Author Response
- Translational Impact in Introduction
- Reviewer Comment: Introduction-Add a sentence clarifying the translational impact: how this framework can directly influence clinical trial or patient management decisions.
- Response: Thank you for your comment. We have added a dedicated statement in the Introduction to clearly describe how our framework links to translation in clinical practice. Specifically, we clarify that the systematic integration of multi-modal biomarkers enables clinicians to more effectively stratify patients for enrollment in clinical trials, informing both trial design and individualized patient management strategies. Page 2, Last Paragraph, Lines 86-97):
“Recent advances in oncology have catalyzed the development of integrative strategies for incorporating biomarkers into clinical decision-making [34-38]. In this review, we propose a comprehensive framework that unifies diverse biomarker categories, including molecular signatures and gut microbiome profiles, to enhance patient stratification and inform therapeutic decision-making. Moreover, by encompassing molecular and genetic profiling, imaging, histopathology, immunohistochemistry, proteomics, metabolomics, lipidomics, and liquid biopsy, this framework collectively provides clinicians with robust and actionable tools to guide clinical trial design, refine patient selection, and personalize treatment regimens (Figure 1). The systematic integration of multimodal biomarkers enables clinicians to more effectively stratify patients for enrollment in clinical trials, informing both trial design and individualized patient management strategies. Thereby, it maximizes therapeutic efficacy and safety while simultaneously addressing the inherent limitations of single-biomarker approaches.”
- Thematic Coherence and Conceptual Flow
- Reviewer Comment: The content reads as a long list without conceptual flow; consider reorganizing for better thematic coherence.
- Response: We have structurally reorganized the manuscript, grouping information into clearly defined sections, and have added transitional paragraphs and conceptual diagrams (see revised Figures and Section subheadings). These changes improve narrative flow and thematic coherence.
- Addition of Specific Clinical Reports and Discussions, Biomarker Detection Table
- Reviewer Comment: Add actual reports/exemplars in each section, especially biomarker
detection. Discussion is general; improve depth with examples/reports.
- Response: Appreciate the reviewer’s insight here. We have created a new table (Table 2) that summarizes the key biomarkers detected by various assays and incorporated it into the manuscript.
- Figure 1 Legend
- Reviewer Comment: Improve the detail and clarity of Figure 1 legend.
Response: Thank you. The legend for Figure 1 has been substantially expanded to ensure it is detailed and self-explanatory.
“Figure 1. Overview of the five major types of immunotherapy in cancer treatment. This schematic illustrates the principal immunotherapeutic strategies currently applied in oncology. (1) Immune checkpoint inhibitors (ICIs): monoclonal antibodies that block inhibitory receptors to restore T-cell activity against tumor cells. (2) Cancer vaccines: agents designed to elicit or enhance tu-mor-specific immune responses by presenting tumor-associated antigens. (3) Immunomodulators: cytokines, adjuvants, or agents that broadly stimulate or regulate immune responses within the tumor microenvironment. (4) Monoclonal antibodies (mAbs): laboratory-engineered antibodies that can directly target tumor antigens, induce antibody-dependent cellular cytotoxicity, or deliver cytotoxic payloads. (5) Cell-based therapies: adoptive cell transfer strategies, including natural killer (NK) cell therapy and chimeric antigen receptor T-cell (CAR-T) therapy, which harness and engineer immune cells for enhanced tumor recognition and eradication.”
- General Information—Need for Specific Examples
- Reviewer Comment: Information is generally described; add specific examples, e.g. in section 3.5, explain mechanisms of SCFA production impact.
- Response: We appreciate the reviewer’s view and have provided a detailed explanation at the molecular and immunologic level of how short-chain fatty acids (SCFAs) derived from the microbiome influence tumorigenesis, referencing mechanistic studies in colorectal and other cancers (Page 10, Lines 378-392):
“The identification of cancer preventive biomarkers is an emerging focus in oncology, supporting strategies to reduce cancer incidence. Short-chain fatty acids (SCFAs), mainly acetate, propionate, and butyrate produced by gut microbiota, are promising candidates [135,136]. Mechanistically, SCFAs exert their effects through several pathways. Butyrate functions as a histone deacetylase (HDAC) inhibitor, leading to hyperacetylation of his-tones and re-expression of tumor suppressor genes such as p21 and p53, which promote cell-cycle arrest and apoptosis [137,138]. SCFAs also engage G-protein–coupled receptors inhibit cell proliferation, induce apoptosis, and cycle arrest via the NF-κB, MAPK, ERK1/2, PI3K, and Wnt signaling pathways [139,140]. Moreover, they can improve the therapeutic efficacy of chemotherapy by enhancing the sensitivity of cancer cells to fer-roptosis [141][142]. In colorectal cancer (CRC), butyrate provides an energy source for normal colonocytes but induces apoptosis in cancerous colonocytes, exploiting the "bu-tyrate paradox" [143,144]. These attributes position SCFAs as potential biomarkers and therapeutic adjuncts in cancer prevention and treatment.”
- Section 5.1 Non-Bacterial Microbiome and Translational Challenges
- Reviewer Comment: Add discussion of non-bacterial microbiome and address translational challenges from models to humans.
- Response: Thank you for this comment. We have expanded Section 5.1 to include non-bacterial constituents (e.g., virome, mycobiome) and provide a nuanced discussion regarding the limitations of animal studies versus human applications. Challenges such as diversity, host-microbial interactions, and microbiome standardization are discussed (Page 11, Lines 447-460):
“Microbiome research now recognizes that viruses, fungi, and protozoa—beyond bacteria—play important roles in cancer biology and therapy [164,165]. For example, fungal signatures across cancer types (e.g., Cladosporium enriched in breast cancer; Aspergillus in lung cancer), suggesting prognostic value [166]. Moreover, fungi interact with bacteria and modulate host immune responses [167]. Viruses such as EBV, HPV, HBV, HCV, and HTLV-1 contribute to cancer through direct oncogenic effects or chronic inflammation, while the broader virome may indirectly influence carcinogenesis by modulating immune development and interacting with bacterial communities [168]. Moreover, the interplay among fungi, viruses, and bacteria plays a critical role in shaping host immune responses and cancer outcomes [167,169]. Incorporating non-bacterial microbiome components into cancer immunotherapy could reveal new therapeutic targets to enhance treatment efficacy and overcome resistance [170-173]. Thus, expanding micro-biome research beyond bacteria is essential for a more comprehensive understanding of tumor biology and for developing next-generation cancer diagnostics and therapies.”
- Figure 4 Clarification
- Reviewer Comment: Clarify “no.4” in Figure 4 – is it clinical applications?
- Response: Many thanks. We have amended Figure 4 and its legend to clearly identify “no.4” as clinical applications of liquid biopsy and related circulating biomarkers
- Reduction of Repetitive Sentences
- Reviewer Comment: Correct repetitive sentences throughout the manuscript.
- Response: We have thoroughly revised the text to eliminate redundancy and repeated concepts, improving overall readability and flow.
- References Consistency and Relevance
- Reviewer Comment: Some references may not align with the content—e.g., section 8.1,
ref 130.
- Response: All references have been checked for relevance and consistency, with incorrect or irrelevant citations replaced appropriately. Reference formatting has also been updated to comply with Biomolecules guidelines.
- Critical Analysis of Challenges and Controversies
- Reviewer Comment: Manuscript lacks critical analysis of challenges, limitations, controversies; technical advances should be contrasted for their sensitivity/specificity.
- Response: Many thanks for this comment. We have added critical commentary to each section, addressing the limitations, unresolved controversies, and clinical trade-offs of emerging technologies. Direct comparisons of sensitivity/specificity of methods are now included where available.
- Minor Comments: Abbreviations, Nomenclature, Figures
- Reviewer Comment: Ensure abbreviations are defined at first mention; maintain nomenclature consistency; improve figure resolution; provide comprehensive abbreviation list.
- Response: abbreviations are defined at first use and a full list is provided.
- All nomenclature has been harmonized throughout the text.
- All figures have been revised for enhanced resolution and clarity.
Reviewer 2 Report
Comments and Suggestions for Authors
In their work the authors presented a Comprehensive Oncological Biomarker Framework, overviewed and systematized existing to date knowledge concerning the cancer predictive biomarkers in personalized oncology. The subject of the manuscript at hand seems to be relevant and topical. The paper under review is validly organized, structured, well written and clear. However, some minor concerns listed below should be addressed by the authors before publication in the Biomolecules journal:
Unfortunately, the authors, having considered multi-omics approaches in detail in their work, did not pay due attention to lipidomics, while a number of papers concerning abnormalities in the lipidome/phospholipidome of glioblastoma, CML, lung, breast, colorectal cancer and some other malignancies have been published over the last decade.
References do not meet the journal's “Instructions for Authors”.
Author Response
Reviewer #2
- Lipidomics Section Expanded
- Reviewer Comment: Insufficient attention paid to lipidomics; many papers on cancer lipidome exist.
- Response: Thank you for this comment. We have added a dedicated subsection on lipidomics, referencing recent studies in glioblastoma, CML, lung, breast, and colorectal cancer. We discuss how integration of lipidomic data enhances framework precision and enables identification of novel biomarkers for cancer stratification (Page 2, Line 93; Page 7, Line 261; Page 17, Line 740). We have also added section 3.5 describing the role of short chain fatty acids (SCFA) and its regulatory action as preventive biomarkers (page 9, lines 372-382).
- References Compliance
- Reviewer Comment: References do not meet journal’s “Instructions for Authors.”
- Response: All references have been reformatted in line with Biomolecules requirements and checked for accuracy.
Round 2
Reviewer 1 Report
Comments and Suggestions for Authors
The authors have addressed most of the previous comments; however, the heading for the last section in Figure 4 is still missing. The authors are advised to review and correct this.
Author Response
Much appreciate your comments and support for our work. We have amended Figure 4 accordingly to reflect all the stages and the respective legend.
Reviewer 2 Report
Comments and Suggestions for Authors
The authors have addressed most of my concerns. The reviewer finds the revised version of the manuscript much improved in term of data presentation and organization and approves it for the publication in the Biomolecules journal.
Author Response
Thank you for your time and effort in reviewing our manuscript.